# MERP: A Music Dataset with Emotion Ratings and Raters’ Profile Information

**DOI:** 10.3390/s23010382

**Published:** 2022-12-29

**Authors:** En Yan Koh, Kin Wai Cheuk, Kwan Yee Heung, Kat R. Agres, Dorien Herremans

**Affiliations:** 1Information Systems Technology and Design Pillar, Singapore University of Technology and Design, Singapore 487372, Singapore; 2Yong Siew Toh Conservatory of Music, National University Singapore, Singapore 117376, Singapore; 3Centre for Music and Health, National University Singapore, Singapore 117376, Singapore

**Keywords:** emotion prediction, music, music emotion dataset, affective computing

## Abstract

Music is capable of conveying many emotions. The level and type of emotion of the music perceived by a listener, however, is highly subjective. In this study, we present the Music Emotion Recognition with Profile information dataset (MERP). This database was collected through Amazon Mechanical Turk (MTurk) and features dynamical valence and arousal ratings of 54 selected full-length songs. The dataset contains music features, as well as user profile information of the annotators. The songs were selected from the Free Music Archive using an innovative method (a Triple Neural Network with the OpenSmile toolkit) to identify 50 songs with the most distinctive emotions. Specifically, the songs were chosen to fully cover the four quadrants of the valence-arousal space. Four additional songs were selected from the DEAM dataset to act as a benchmark in this study and filter out low quality ratings. A total of 452 participants participated in annotating the dataset, with 277 participants remaining after thoroughly cleaning the dataset. Their demographic information, listening preferences, and musical background were recorded. We offer an extensive analysis of the resulting dataset, together with a baseline emotion prediction model based on a fully connected model and an LSTM model, for our newly proposed MERP dataset.

## 1. Introduction

With the explosive growth in the amount of music available online, developments in the field of Music Information Retrieval (MIR), such as models to classify and analyse music, have become ever more important [1]. One of the MIR tasks that has gained increasing attention is the automatic recognition of emotions from music, or Music Emotion Retrieval (MER). The field of MER focuses on constructing statistical, machine learning models that can predict perceived emotion based on music audio. This field has grown rapidly in the last decade or so, partly due to the growth of the music industry in the digital space, which makes it easier for researchers to access large datasets of music. Because emotion is subjective and often vaguely defined in studies [2], different listeners may have differing views on the emotion they perceive from a song. This results in noisy emotion labels in music datasets. In order to attempt to reduce this noise, our research explores whether we can find reasons for the difference between listeners. We thus explore whether similarities between listeners translate to similarities in the perception of emotion in music. Identifying such similarities would contribute to the topic of personalizing dynamic MER, allowing models to be customised to individuals. With this objective in mind, we present a newly gathered large open-source dataset (https://www.kaggle.com/kohenyan/music-emotion-recognition-with-profile-information (accessed on 26 December 2022)) of music labelled with emotion ratings, as well as profile information about the raters (e.g., gender, musical preferences, etc.). We refer to this dataset as MERP, which stands for Music Emotion Recognition with Profile information (MERP). The MERP dataset is available online, together with baseline models for emotion prediction (https://github.com/dorienh/MERP) (both with and without using listener profile information (accessed on 26 December 2022)).

Despite the surge of interest in MER, the number of datasets with emotion annotations is limited, especially those with dynamic annotations, i.e., ratings that are collected continuously throughout the piece. To the authors’ knowledge, the DEAM dataset (1802 fragments by 15–32 annotators) [3], the Moodswings dataset (240 fragments by 7–23 annotators) [4], and the MuVi dataset (81 fragments by 5–9 annotators) [5] are the only datasets that offer music with dynamic emotion (arousal and valence) annotations. For a more complete overview of existing music datasets with emotion annotations, the reader is referred to Chua et al. [5]. In order to train any powerful machine learning model, the availability of big datasets is crutial. This was an important motivation for creating a new dataset for full-length (creative commons) music pieces with high-quality, dynamic emotion (valence-arousal) annotations.

There are many individual differences in music perception stemming from the listener’s musical background, genre preferences, and more. Identifying listener features that influence affect perception may potentially improve MER for different listener groups. Hence, in this work, we present the MERP dataset, which is catered towards exploring whether MER can be improved given additional listener profile information. The dataset contains copyright-free, full-length musical tracks with dynamic ratings on Russell’s two-dimensional circumplex model [6], as described in Section 2.1. To our knowledge, this is the first work to present a publicly available dataset of dynamic affect labels of full-length musical pieces, alongside profile information of participants.

In the remainder of this manuscript, we provide a brief background of emotion models, and the effect of profile features on emotion perception (Section 2). This is followed by a thorough description of the dataset collection process (in Section 3), followed by a statistical analysis and visualisations in Section 4. Extensive preprocessing and denoising was performed to increase the quality of our data. In Section 5, we use the resulting dataset to train a baseline emotion prediction model and evaluate the influence of the different profile features. The benchmark results for this dataset are listed in Section 6, followed by the conclusion.

## 2. Related Work

We provide a brief overview of related work that informed some of the decisions made while carrying out this study. We start by exploring different models of emotions, and then move on the how different profile features may influence emotion perception.

### 2.1. Categorical versus Dimensional Models of Emotion

The relationship between music and emotions has been scientifically explored for at least a hundred years (e.g., Seashore [7]), with a surge of interest beginning in the 1950s by Meyer [8], and expanding even more widely in recent decades, both in music psychology [9,10,11] and Music Information Retrieval (MIR) [5,12,13,14]. Generally, there are two main ways of capturing and representing emotion in music: categorical and dimensional [15]. Representations that use discrete emotional terms fall into the former type. Music datasets that use this type of emotion representation include the CAL500 dataset [16], which provides a three-scale rating of eighteen emotions for each song, and the Emotify dataset [17], which classifies each song into one of nine categories of the Geneva Emotional Music Scales [18]. As summarized by Barthet et al. [19], many different types of emotion models with categorical labels exist. Some studies use discrete terms directly [20], while other studies propose clusters or groups of discrete emotional terms. For instance, Trohidis et al. [21] propose twelve emotion clusters, while Hu and Downie [22] propose five clusters for the Audio Mood Classification task of the annual Music Information Retrieval Evaluation eXchange (MIREX). Dimensional models, on the other hand, attempt to abstract the representation of all emotions along two or more dimensions. A widely known two-dimensional model of emotion is Russell’s circumplex model of affect [6]. In this model, valence (V) represents the positivity or negativity of emotion (from unpleasant to pleasant), and arousal (A) refers to the intensity/energy level of the emotion (from low to high). In this way, all emotions can be represented using these two dimensions of V and A. The Lakh-Spotify Dataset [23] is one of the latest datasets that uses symbolic music paired with emotion labels in terms of VA. Valence and Arousal labels have also been used for tasks such as controlling emotion in generated music [24,25,26,27] as well as variation detection in emotion from music [28]. Due to the nature of the two representations, MER techniques for analyzing categorical annotations usually involve classification, while dimensional annotations require regression techniques.

We should note that categorical and dimensional emotion representation models are closely related, and dimensional representations are often utilised in a categorical manner. For example, Bischoff et al. [29] divide Thayer’s two-dimensional Energy and Tension model [30] into four quadrants of the two-dimensional plane, while Han et al. [31] section the two-dimensional model into eleven subdivisions, where each sub-dimension is represented by a discrete emotional term. Soundtracks [32] is a dataset that collected both categorical and dimensional annotations, and compared the two representations of emotion. Their results show that the perceived emotional labels collected through both representations are largely comparable for the Soundtracks dataset. More recently, [25] provided a method for mapping discrete emotional terms onto Russell’s dimensional model. Finally, Chua et al. [5]’s MuVi dataset provides both dynamic ratings along the valence and arousal dimensions throughout the song, as well as static emotion labels (categorical labels for the entire song) for music and video stimuli.

Although these two types of emotion representations are similar, dimensional representations are more versatile in two ways. Firstly, due to the continuous nature of dimensional representations, they are able to represent different degrees of emotion [33], while categorical representations do not typically capture the degree of emotion (with some exceptions). Secondly, with dimensional representations, it is more practical to represent changes in the emotion of music over time [15,34]. Therefore, we see that categorical emotion labels are often used for making static annotations, i.e., a single annotation for a song or excerpt. Dimensional representations, on the other hand, are better suited to capture changes in emotion throughout a song. The DEAM dataset [3], which helped inspire this work, contains both static as well as dynamic dimensional emotion annotations. In this work, we use valence and arousal annotations to capture how perceived emotion evolves dynamically over time throughout the length of entire songs. This design decision was made because dynamic annotations capture the dynamic nature of music and the temporal evolution of emotional content. Furthermore, dynamic labels can be aggregated to create static labels when needed.

### 2.2. Impact of Listener Demographics on Perceived Emotion

Emotion perception during music listening can depend heavily on listener characteristics [35]. Pearce and Halpern [36] and Lima and Castro [37] both found similarities and differences in emotion perception of music among older and younger adults. They observed that the extent of sadness perceived decreases as age increases. Musical training was also reported to have an impact on emotion perception. In addition, lower frequencies were rated with lower valence by musicians in a study by [38]. This finding may be due to the impact of musical training on one’s perception of musical cues and their relation to conveyed emotion [39]. Moreover, Schedl et al. [40] found higher agreement in labels between participants with musical training and those who play an instrument compared with those lacking training. Lima and Castro [37] similarly observed a correlation between number of years of musical training and accuracy of music emotion categorization.

Another listener feature that may have an impact on perceived emotion is culture: while listeners are generally able to accurately identify emotion in music from cultures foreign to them [41,42], cultural background has been reported to impact the participant’s perceived emotion *agreement* of music. A comparison study between a Greek group of participants and a group with varying cultures reported that participant agreement was higher in the Greek group [43]. Lee et al. [42] found that participants from Brazil, South Korea, and the US mostly agreed when recognising simple emotional characteristics, but showed disagreement when recognising more complex emotional characteristics such as dreamy and love. Stereotypes of a culture may also have an impact on one’s perception of music from specific cultures [44]. Wang et al. [45] compared the impact of musical background and cultural background on the perception of emotions by Chinese and Western participants. They found cultural background to have a larger impact as compared to the musical background of a participant. Such findings suggest that people from the same cultural background may show higher levels of agreement with regard to perceived emotion. Music from different cultures has also been shown to convey emotion differently through various musical features [46]. Chen et al. [47] managed to improve the quality of a music valence prediction model by taking cultural differences in music features into account. Due to the importance of all these listener features, we included a large number of profile features in our dataset. In the next section we will detail the procedures followed to create our new dataset.

## 3. Data Gathering Procedure

To create a new dataset, we could approximate a categorically labelled dataset by crawling the web and finding emotion tags on resources such as Last.fm and AllMusic, as done in [48,49,50], however, this would not provide us with curated, dynamic data. Datasets with dynamic valence and arousal labels are typically collected manually. Datasets such as DEAP [51] and PMEmo [52] go one step further and also include physiological signals, which requires participants to be present physically. Alternatively, in an effort to collect larger quantities of affect labels in a shorter amount of time, although with a potential loss in accuracy, crowd-sourcing on platforms such as Amazon Mechanical Turk (MTurk) has also been explored [3,53,54,55,56]. Some researchers utilize a mix of both online and offline collection methods [57,58], or even use predictive models such as AttendAffectNet [59] for the emotion labeling [60]. Regardless of the data collection method, it is important for each musical excerpt in the dataset to be labelled by multiple participants in order to account for subjectivity. Participant agreement can be used to identify anomalies in the labels, or be aggregated to better represent the general response.

We opted for large-scale collection in this work, and used the MTurk platform. Given the noise that often comes with this collection method, extra attention was put on preprocessing the data and filtering out noisy annotations, as is explained in detail in the remainder of the section. Additionally, each participant received four stimuli previously labelled by an expert in the DEAM dataset. This offers us a benchmark to filter out low-quality annotations.

### 3.1. Participants

We collected data through Amazon Mechanical Turk (MTurk). The listening study was carried out online, as a Human Intelligent Task (HIT) on the platform. There are two types of participants, ‘master’ and ‘non-master’ participants. Master participants are generally more reliable, as they have to go through a screening process to prove their reliability before earning the master title. A total of 452 participants completed the task on MTurk, of which 171 were master participants and 286 non-master participants.

At the beginning of the listening study, profile information of the participants was collected through 9 brief questions. With the data collection platform in mind, the questions were set to be factual and easy to answer. They also served to check the attentiveness of participants [61]. The questions can be categorised into 3 sections: demographic information, listening preferences, and musical experience. For demographic information, participants were asked about their age, gender, country of residence, and country of musical enculturation. For listening preferences, we asked them what language of music they like to listen to most, as well as their favourite genre of music. As for musical experience, they were asked if they were actively playing at least one instrument, whether they have received formal training, and if they have, how many years of musical training they received. We describe the groups of participants for each profile in Section 4.2, and explore whether the profile information of participants is helpful in an automatic music emotion recognition task in Section 5.

Using Amazon Mechanical Turk (MTurk) allowed us to access a large pool of participants from different continents in a quick and convenient way. To ensure quality, we applied a novel technique of having participants rate benchmark songs so we could do basic filtering, which we enhanced with other preprocessing techniques as described in Section 4.1.

### 3.2. Stimuli

To ensure that the collated dataset would be publicly accessible, the stimuli were selected from readily available Creative Commons sources, namely the Free Music Archive (FMA) [62] and the Database for Emotional Analysis in Music (DEAM) [3].

The FMA is a large database that consists of 106,574 full-length tracks. From the FMA all-time chart, we looked at the top 1000 songs listened to, and filtered out the songs shorter than 30 s and longer than 10 min. Due to budget constraints for the study, we could not annotate all of these songs and we further narrowed the selection. To do this, we used an existing, trained emotion prediction model [63] and selected songs which had the most distinctive emotion. To do this, we extracted features using the OpenSmile toolkit [64], which were then fed into a Triple Neural Network, trained as described in [63], to determine a static (single) arousal value and valence value for each song. The valence-arousal values of the songs are plotted in Figure 1, where we can see that the distribution of the songs is relatively sparse in the high arousal/low valence quarter, as well as the high valence/low arousal quarter. This is not unexpected, as the valence and arousal of music are often related. For example, a sad song is often slow and low in energy, which would make it low in both arousal and valence. To ensure that the stimuli represent emotions from all quadrants of the valence arousal graph, we first binned both valence and arousal of each song into 5 categories. The categories are low valence and low arousal; low valence and high arousal; mid valence and mid arousal; high valence and low arousal; and high valence and high arousal. For each of these 5 categories, we selected 10 songs that were the most distinct from other categories, resulting in the final 50 songs that were used in our study. Figure 1 highlights the 50 songs selected as stimuli, visualised on the valence and arousal graph.

In addition, 4 song excerpts (each 45 s in length) were selected from DEAM. These excerpts also originate from the FMA dataset, and have annotated (dynamic) valence arousal values. These songs were presented to every participant, hence the ratings for these songs could be used as a quality benchmark to filter out noisy entries during the data preprocessing stage (see Section 4.1). The total length of the 54 songs selected is about 8778 s, which is approximately 146 min, or 2 h and 26 min. The shortest song is around 31 s, while the longest song is 4 min and 58 s, and the mean length is 2 min and 52 s. Table 1 shows the total duration across the 54 music stimuli based on their Valence/Arousal category.

### 3.3. Procedure

The listening study was organised on a single web page on Amazon MTurk, in which all questions were listed and could be scrolled through freely by participants during the answering process. The participants were first asked a series of 9 questions about themselves. They were then introduced to the task, as well as the definitions of valence and arousal, along with examples of songs that have high and low arousal and valence. Note that in this work, we focus on *perceived* emotions, i.e., the emotions the listener perceives as being expressed by the music. The examples were accompanied by a 2-dimensional valence/arousal graph, with a dot that marks where the example piece is positioned on the VA graph. Participants were hence provided with examples of the graphical representation of emotion in music so that they become familiar with the meaning of valence and arousal.

A total of 24 songs were presented to each participant, of which, 4 songs were the benchmark DEAM songs which were presented to all of the participants. The remaining 20 songs were randomly selected from our 50 stimuli. The 4 DEAM songs were presented in the task as the 1st, 4th, 7th and 10th stimuli, though not in the same order. A very generous 2 h were set as a maximum duration for this listening task, while participants were told that the study should take about 30 min.

During the listening study, the participants were required to label VA values simultaneously while listening to the stimuli in real-time. We captured their mouse position over a two-dimensional VA graph throughout a song; this is unlike some other studies where participants labelled Valence and Arousal separately [3], listened to the song multiple times [58] or were allowed to edit their answers [2]. In this way, we were able to capture the initial impressions of the participants while keeping them constantly engaged in the labelling task.

Han et al. [65] shows that dynamic tracking in music emotion recognition is more in line with the characteristics of music than static processing. For participants to familiarise themselves with the valence arousal graphical labelling interface (Figure 2), which dynamically tracks the user’s mouse position after they press play on a song, a practice song is first provided prior to the actual questions. Clicking on the centre of the graph will begin the streaming of the music, as well as the recording of their mouse position on the graph. Subsequently, for each stimulus, a valence arousal graphical labelling interface is displayed, along with reminders of the definition of valence and arousal, and of the instructions (to constantly indicate their perceived emotion in valence and arousal throughout the song). As full task completion was not obligatory, on average, each participant labelled 13.8 songs. The javascript code of the developed rating interface that integrates with MTurk is available online (https://github.com/dorienh/MERP/blob/master/amazon_Merged.html (accessed on 26 December 2022)). This interface samples the participants’ mouse position at a frequency of 10Hz to collect the valence and arousal values.

## 4. Dataset Analysis and Visualisation

### 4.1. Data Filtering

Data collected through online crowdsourcing methods is generally known to be noisy [66], especially so when collecting subjective data. Multiple methods of filtering were carried out to identify (and remove) entries of low integrity from the collected dataset.

#### 4.1.1. Step 1—Identifying Erroneous Entries from Profile Information

We first began with a screening of the participants. Of the initial 452 participants, 5 of them did not complete the task and were removed from the dataset, leaving a total of 447 after this initial filtering. Even though participants were instructed to only complete the task once, due to releasing the task in multiple batches, 26 participants submitted entries in more than 1 batch. 20 out of these 26 participants submitted conflicting entries when answering the profile questions in the first half of the task, and were removed from the dataset. Submissions by participants who made obvious mistakes while answering profile questions in the first half of the task were also disregarded. For instance, some participants had negative numbers or numbers close to 2000 when asked how many years of formal music training they had received. Other participants indicated that they had not received formal music training, yet entered a positive number for the number of years they had received training. As a result, a total number of 417 participants remained in the dataset after this step.

#### 4.1.2. Step 2—Discarding Trials with Abnormal Length

Due to inconsistent sampling frequency caused by variables such as network connection, system latency, browser used, and CPU usage [54], some of the collected emotion ratings for the same songs were of slightly varying lengths. For example, a 30 s excerpt should have 300 labels but instead had too few, or too many. Trials that were too short were discarded, to avoid data fabrication. For trials that were too long, a threshold of a difference of 20 data points (2 s) was determined to be of acceptable distance from the length of the audio features extracted from the songs. Any additional ratings after the song ended (max. 20) were removed. After this operation, some of the ratings of the 4 songs from the DEAM dataset, which are meant to provide us with a common benchmark amongst all participants, would have been removed. To avoid this, we further removed participants who had a rating for a DEAM song shortened. This left us with 358 participants and 4441 trials.

#### 4.1.3. Step 3—Discarding Entries That Greatly Differ from DEAM

To determine the integrity of the affect labels collected, we compared them with the labels provided in the DEAM dataset. The labels in the DEAM dataset were averaged across participants, and for each time step, we checked whether our collected annotations fell within 2 standard deviations of the average DEAM label. Songs whereby more than 50% of the time steps were within this threshold, were considered to be acceptable. All 4 DEAM songs for each participant had to fulfil this check, or the participant was discarded. After this procedure, we were left with 277 participants and 3482 trials, as displayed in Table 2.

#### 4.1.4. Resulting Dataset

An overview of the resulting data (before and after preprocessing) collected through MTurk for both master as well as non-master participants, is shown in Table 2. Each rated stimulus is considered a trial.

### 4.2. Profile Visualisations

One of the novel aspects of this dataset is the inclusion of profile information that was gathered about the raters. In this section, we visualize the demographics of the 277 participants. In Figure 3, we can see an overview of the proportion of participants that fall into each profile category. The profile information was binned as seen in the figure. This binning will be useful in the later sections on emotion prediction models. Section 5.1 further describes how the profile information is treated and utilised for emotion prediction.

The first 3 feature bars of Figure 3 show common demographic profile features, e.g., age. As depicted in Figure 4, most of the participants are young adults in their twenties to thirties. The age feature was divided into 4 bins. Participants below 25 years of age are considered youth, between 26 and 35 years of age are young adults, between 36 and 50 years of age are adults and above 51 are elders. After binning, 52.7% of participants are youth, 24.2% are young adults, 13.4% are adults and 9.7% are elders. The second bar in Figure 3 shows that there are 57.0% male and 43.0% female participants. The third bar depicts the country of residence of participants. The majority of the participants are from the USA (52.7%) and India (42.6%) as the MTurk task was released to these two countries. The remaining 4.7% includes participants from Great Britain (1.4%), Italy (0.7%), South Africa (0.4%), Russia (0.4%), Indonesia (0.4%), Armenia (0.4%), American Samoa (0.4%), Romania (0.4%) and Brazil (0.4%).

The fourth feature bar, labelled ‘enculturation’ in Figure 3, depicts a slightly more unique feature which represents the musical enculturation of participants—which country’s music do participants identify with most. As one might expect, the division looks very similar to the bar above it, implying that country of residence and country of musical enculturation are related. The percentages of each country are as follows: USA (52.7%), India (40.8%), Great Britain (1.8%), Italy (0.7%), Japan (0.4%), Ecuador (0.4%), Mexico (0.4%), South Africa (0.4%), Russia (0.4%), Armenia (0.4%), Colombia (0.4%), American Samoa (0.4%), New Zealand (0.4%), United Arab Emirates (0.4%) and Brazil (0.4%).

The fifth and sixth feature bars of Figure 3 pertain to the listening preferences of participants. With regard to the preferred language of lyrics, since participants are mostly from the USA and India, it is unsurprising that for the fifth feature ‘language’, songs with English (72.2%) and Tamil (18.1%) lyrics are the favourite of most participants. The remaining 9.7% include the languages Malayalam (3.2%), Hindi (2.5%), Italian (0.7%), Telugu (0.7%), Armenian (0.7%), Japanese (0.7%), Korean (0.4%), German (0.4%) and Bengali (0.4%). As for the preferred genre of participants shown in the sixth bar named ‘genre’, Rock (31.8%) had the highest percentage, followed by Classical (14.1%) and Pop (13.7%) music. Many other genres were grouped as ’other’ in the bar chart as they were small in comparison, they include Rhythm and Blues (8.3%), Indie Rock (6.9%), Country (6.9%), Jazz (5.4%), Electronic dance music (2.2%), Metal (2.2%), Electro (1.4%), Techno (1.1%) and Dubstep (0.7%).

The seventh to ninth feature bars in Figure 3 represent the musical experience of participants. The seventh feature, labelled ‘instrument’, represents the proportion of participants that are actively playing at least one instrument. 45.8% of them indicated that they were actively playing an instrument. The eighth bar, named ‘training’, depicts the proportion of participants that have received formal musical training. A total of 57.4% of participants indicated that they received formal musical training, while 42.6% indicated that they never received formal music training. Since 42.6% of participants have not received formal musical training yet 45.8% are actively playing an instrument, we can surmise that at least 3.2% of participants are self-taught. The ninth bar, labelled ‘duration’, reflects the number of years participants have received formal training. The 42.6% of participants who had not received formal training are included in the ninth feature bar as the participants who have received 0 years of training. A total of 5.8% participants underwent 1 year of training, 15.9% had 2 years of training, 13.0% had 3 years, 5.8% had 4 years, and 7.9% had 5 years of training. Overall, 48.4% of participants received between 1 and 5 years of formal musical training. 0.9% of participants indicated having 6 or more years of training, the largest value being 31 years of training.

In the tenth feature bar of Figure 3, the proportion of MTurk master to non-master participants is represented. A total of 46.2% of participants are master participants while 53.8% of participants are non-master participants. It is noteworthy that 128 master participants were retained from the original 172 master participants after our preprocessing, while only 149 non-master participants were retained from the original 280 non-master participants. The retention percentage of 74.4% for master participants as compared to only 53.2% for non-master participants implies that master participants are indeed more reliable compared to non-master participants.

We should note that the profile binning or grouping for non-boolean type profiles was arbitrarily determined in this work. For example, as the age of participants was largely skewed towards the young adult age, the two younger groups are of smaller age ranges while the two older groups are of larger age ranges. In future research, one could experiment with different configurations, or further testing may be performed to determine more representative age bins that show a difference in perceived emotion from the music. The same can be said for the preferred genre profile type. In this study, the participants mainly preferred rock and classical songs. Some of the favored genres were not represented by many participants, and were grouped under ’Other’. Perhaps with better representation, more significant differences between genres would be revealed.

### 4.3. Statistical Differences in Affect Ratings between Profile Groups

We analysed the collected data in order to determine whether there are significant differences in terms of valence and arousal annotations from participants of various demographic groups. As statistical testing requires independent samples, the dynamic affect labels were averaged to a single value per participant per song. Additionally, because the valence and arousal ratings were not normally distributed, a non-parametric test was used. The non-parametric Kruskal Wallis test [67] was used for each of the 10 profile features to identify whether statistically significant differences exist between the emotion ratings of the different profile groups. Table 3 shows the results of the Kruskal Wallis tests. *p*-values lower than the threshold value of 0.05 are marked in bold so as to highlight that there is a significant difference in emotion ratings between profile groups of that profile feature.

For the profile and affect type pairs that have *p*-values below 0.05, we carried out Dunn’s test [68] as a post hoc test, with Bonferroni correction [69]. The resulting *p*-values of the Dunn’s test indicate which profile features are statistically different. For each bold value in Table 3, we report the profile features that are significantly different, along with their *p*-values below. Our findings are in line with Schedl et al. [40], who found differences in music perception only for some user groups.

A statistical difference was found between valence ratings provided by young and adult raters (p=0.0484) as well as youth and elder raters (p=0.0291). The data suggests that the youth group tends to give higher valence ratings as compared to the two other groups. Valence ratings from the young-adult age group seem to lie in between the other groups, suggesting that the perceived valence of music may decrease with age.

Both valence and arousal ratings from raters from a different country of residence showed a significant difference. For valence, however, the post hoc test *p*-values were larger than 0.05 after Bonferroni correction. In particular, between the USA and India participants, the *p*-value was 0.0560, which is close to the threshold for significance. As for arousal, a significant difference was found between USA and India (p=0.0167). Participants residing in India had a larger proportion of ratings that were near the origin (0,0), for both affect types, as compared to participants residing in the USA. In general, the ratings from participants residing in the USA were more evenly spread out as well, while ratings from participants residing in India were skewed towards the positive end of both affect types.

With regard to raters with a different country of music enculturation, a significant difference between the USA and India groups was found for both valence (p=0.0076) and arousal (p=0.0484), and between the USA and other countries, only for valence (p=0.00004). It is worth noting that there is only one participant representing the ‘other’ group, hence we did not take this value into consideration for the analysis. Furthermore, as there is a larger overlap between the participant groups for country of residence and country of music enculturation, similar observations of the data can be made.

For listeners with a different preferred genre of music, we see a statistically significant difference in terms of valence ratings. Interestingly, the differences are between the classical genre and each of the other groups. Namely, between classical and rock (p=0.0004), classical and pop (p=0.0766), and classical and other (p=0.0001). As compared to the other three genres, participants who prefer classical music mostly rated valence closer to the origin. Other groups tended to give higher positive valence ratings.

Participants who actively play an instrument compared to participants who do not, have a larger proportion of ratings near the origin (0,0). With regard to valence (p=0.0002), participants who do not actively play an instrument had the most ratings near 0.5 valence. As for arousal (p=0.0383), other than the larger proportion of ratings near the origin by participants who do not actively play an instrument, both groups are generally skewed towards more ratings in the positive arousal quadrant rather than the negative quadrant.

The distributions for participants who have received formal training and those who have not (the eighth profile information-training), closely resemble those of participants who actively play an instrument and those who do not (the seventh profile information-instrument). This is observed despite the fact that there are 43 participants who have received formal musical training but are not actively playing an instrument, and another 11 participants who are actively playing an instrument but have not received formal training. A statistically significant difference is found between these two groups, for both valence (p=0.0009) and arousal (p=0.0314). This makes sense, as most participants who play an instrument learned to do so through formal musical training.

The group of participants who have not received formal training coincides with the group of participants who have received 0 years of musical training. With regard to arousal ratings, the group of participants with 1 to 5 years of training is significantly different to the other two groups: 0 years (p=0.0145) and more than 5 years of training (p=0.0171). As for valence, a significant difference was found between the group of participants with 1 to 5 years of training and the group with 0 years of training (p=0.0024). The lack of significant difference between the group of 0 years and the group of more than 5 years of training suggests that perhaps the length of duration of training may not have an obvious impact on the perceived affect. The statistical difference noted in both affect types may be due to the large proportion of ratings near the origin, given by the group with 1 to 5 years of training, and not found in the other two groups.

Lastly, the ratings of master MTurk participants showed a statistical difference with non-master MTurk participants where arousal is concerned (p=0.0015). Non-master participants had a large proportion of ratings near the origin, while master participants showed a tendency to rate with higher arousal values. Though the same is observed in valence, the difference between the two groups is not substantial enough to be significant. This peak of values near the origin is observed in many of the profile types aforementioned, which suggests that perhaps those groups have more non-master participants. This is found to be the case for country of residence and enculturation, where approximately 73% of the India group are also non-master. It is also possible that there is a subset of non-master participants that cause this peak. Alternatively, since the mouse pointer is positioned at the origin when the experiment begins, it is possible that non-master participants move their mouse less, or respond later.

The significant differences found above confirm the importance of capturing profile information in a dataset of valence and arousal ratings of music. In the next section, we predict valence and arousal ratings from the audio and profile information captured in this newly proposed dataset, and thus provide a baseline model. The significant differences found between various groups for the different profile types suggest that affect prediction may be improved and refined by feeding the model this information; this is what we will test in our experiments.

## 5. Emotion Prediction Models

In this section, we provide a baseline prediction model for valence and arousal. We provide baseline results for models that use audio features only, as well as models that use both audio features and profile information of participants. We explore two types of model architectures for our music emotion prediction tasks: a fully connected model, and a long short-term memory (LSTM) model. The models are simple in design and intended to be supplementary performance benchmarks on the dataset. In future work, more state-of-the-art methods such as convolutional neural networks [70,71,72,73,74,75,76], or transformer architectures [60,77,78] could be used with the dataset for further experimentation with profile information and its uses for building improved MER models.

### 5.1. Feature Extraction and Label Aggregation

We use two types of features to build our emotion prediction baseline models: audio-based features, and profile features (of the raters).

#### 5.1.1. Audio Features

We extracted audio features from the audio files using the openSMILE toolbox [64]. We used the openSMILE configuration file from the 2015 ‘Emotion in Music’ task from the MediaEval Multimedia Evaluation Campaign. Following existing literature [54,79], a total of 260 low-level features, often referred to as the IS13 acoustic feature set (shown in Table 4) were extracted for every 500 ms segment with frame size 60 ms and step size 10 ms, resulting in a vector of features for every 0.5 s of music [3].

#### 5.1.2. Profile Features

The profile features were binned into categories, such that they are numerically represented with values ranging from 0 to 1, each bin represented by a float value. In a binary example, such as whether a participant actively plays an instrument, ‘no’ is represented by 0 and ‘yes’ is represented by 1. This way, the participant’s profile information is classified into bins and represented by a number. We chose to build a separate model for each profile feature so that we can more clearly identify and gain insight into which profile features are useful to improve prediction accuracy.

#### 5.1.3. Label Averaging over Participants

After collecting labels through MTurk, we have multiple ratings per song. In our final training and test dataset for predicting emotions, we want to have one value per song. We therefore averaged the labels for all (types of) raters per song. More specifically, in the case where we do not consider profile information, all labels for a song are averaged, as described by Equation (Equation 1). This results in 15,849 values for both arousal and valence. Let $Y_{j,i,t}^{\mu }$ be the label rated by participant *j* for song *i* at time *t* and $S_{j}$ is the list of songs that have been labelled by participant *j*. At each time *t*,
(1)$${\bar{Y}}_{i,t}=\frac{\sum_{j}Y_{j,i,t}{𝟙}_{\left\{i\in S_{j}\right\}}}{\sum_{j}{𝟙}_{\left\{i\in S_{j}\right\}}}$$

When taking profile information into consideration in a model, we only average the labels given by users with the same profile feature, per song. For example, when considering the age of participants, since we binned age into 5 categories, we average the labels labelled by participants that fall within each of these 5 categories. As shown in Equation (Equation 2), for a single bin of a single profile type, let *P* represent a profile type (e.g., age, genre), while $P_{r}$ represents all participants that belong to a particular bin of the profile type (e.g., female). Then, ${\bar{Y}}_{i,t,P_{r}}$ would be the averaged label at time step *t* for song *i* by each participant *j* that belongs to $P_{r}$.
(2)$${\bar{Y}}_{i,t,P_{r}}=\frac{\sum_{j\in P_{r}}Y_{j,i,t}{𝟙}_{\left\{i\in S_{j}\right\}}}{\sum_{j\in P_{r}}{𝟙}_{\left\{i\in S_{j}\right\}}}$$

### 5.2. Baseline Emotion Prediction Models

As previously mentioned, two types of models were trained using the newly proposed dataset, a fully connected model and an LSTM model. The purpose of this is to provide a simple benchmark for future research as well as provide source code for the data pipeline so that others can easily use the provided dataset. The two proposed models were each trained on two variations of our dataset: one using audio features alone (averaged per song), and one using audio features concatenated with a single profile type feature (averaged per song and per profile type). This is done separately for arousal and valence. The input to both types of models is 30 timesteps long and contains 15 consecutive seconds of music. As depicted in Figure 5, for models trained using audio features only, the input is straightforward, a vector of shape 260×30. For models trained using both audio features and profile features, we append the profile type feature to each feature vector in each time step, resulting in an input of shape 261×30. Figure 6 shows the two architectures used.

All models were trained using 5-fold cross-validation, where each fold was trained for a fixed number of 100 epochs. Mean squared error (MSE) was used as the objective function, and the Adam optimizer [81] was used with a learning rate of 0.0001. As there are a total of 54 songs, 10–11 songs were withheld as the test set for each fold, while the remaining 44–43 songs were used as the training set. In this way, songs in each training and test set are independent of one another. A batch size of 8 was used.

#### 5.2.1. Fully Connected Model

A dense neural network, i.e., a fully connected model, was implemented as a simple baseline model to show the performance of valence and arousal prediction. This seemingly simple architecture has previously managed to outperform other memory-based models on the task of emotion recognition [82]. We want to provide a very simple benchmark here, as a starting point for other researchers.

Details of the fully connected model architecture are shown on the left of Figure 6. The neural network consists of 3 fully connected layers (512-256-1 nodes) with a dropout of 50% and with leakyReLu of 0.1 as activation function in between consecutive fully connected layers. The first fully connected layer expands the input dimension from 260 to 512, before reducing it by half to 256, then finally to 1. A tanh activation function is then applied to the output, which results in the predicted valence or arousal value. We opted to use the tanh activation function as the valence and arousal values are in the range −1 to 1. To reduce the noisiness of the predictions, a Gaussian kernel with sigma set to 1.5 and size 7 was applied as a smoothing function. The architecture of this fully connected model was inspired by Thao et al. [82], but the hyperparameters were tuned to our dataset using trial-and-error. The source code of the model’s implementation in PyTorch is provided online (https://github.com/dorienh/MERP (accessed on 26 December 2022)).

#### 5.2.2. LSTM Model

Recurrent neural network architectures (RNNs) are typically a popular choice when it comes to sequential input data such as music, video, and speech. LSTMs are a type of Recurrent Neural Network (RNN) that were designed to overcome the vanishing gradient issue. The LSTM cells consist of an input gate, an output gate and a forget gate [83]. This enables the network to keep a memory of previous time steps which is carried forward when looking at subsequent time steps. The forget gate learns to discard unnecessary data. Recurrent neural networks, or more specifically LSTM architectures have successfully been applied to the task of emotion prediction from music [73,83,84,85].

Our baseline model was inspired by the work of [82], enhanced with the idea of bidirectional LSTM layers (bi-LSTM) by [73]. These special layers basically consist of two LSTMs: one that takes input in a forward direction, and one that does so in a backwards direction. On the right side of Figure 6, we see that the first layer has a hidden dimension of 512, doubling the output to accommodate both forward and backward directions. A fully connected layer is then used to reduce the hidden dimension of size 2048 to 1, after which a tanh activation function is applied. The resulting output represents either a predicted valence or arousal value, depending on the type of model trained. The architecture of this bi-LSTM model was inspired by Thao et al. [82] and Jia [73], but the hyperparameters were tuned to our dataset using trial-and-error. The source code of the model’s implementation in PyTorch is provided online (https://github.com/dorienh/MERP (26 December 2022)).

In the results section, we compare the two models to see how the fully connected model fares in comparison to the LSTM model.

## 6. Prediction Results

### 6.1. Emotion Prediction Models Using Audio Only

The VA (valence and arousal) prediction results for the fully connected model are shown in Table 5. We show both the MSE and the Pearson correlation coefficient (R) between the predicted and ground truth VA values as evaluation metrics using 5-fold cross-validation.

From the table, we can see that the fully connected model outperforms the LSTM model in terms of MSE. The MSE values for both valence and arousal are smaller for the fully connected model, while the R-values are larger. A larger R-value indicates that the prediction approximates the ground truth better. Since we are using a very strong feature representation (OpenSmile), it may suffice to use a simple model to make predictions. This is in line with the conclusions of Thao et al. [82], whose fully connected models also outperformed the LSTM model for emotion prediction. They speculate that the 400ms input windows already contain enough temporal information to make an informed prediction of emotion. In our case, since we are using a 500ms window, this may provide an explanation for the better performance of the simpler model without memory. Another possible reason why the LSTM model did not perform as well as the fully connected model could be due to the much larger number of parameters being trained in the LSTM model as compared to the fully connected model. This makes it harder to train without a large dataset, and more prone to overfitting. For the purpose of this study, we do not aim to find the best state-of-the-art performance, but merely offer a dataset with a model pipeline that is made available online (https://github.com/dorienh/MERP (accessed on 26 December 2022)), and that can be used as a benchmark for future research. It is worth noting, however, that the arousal prediction model performs better. This is in line with many other studies [82], and most likely due to the fact that arousal is an easier-to-understand attribute, reflective of the energy of the music.

Given that this is a new dataset, we offer a new benchmark, and cannot directly compare with existing work. Looking at related papers, however, we notice that our results are in the same ballpark range as them. For instance, Aljanaki et al. [3] provide an overview of the best performing models (from 21 teams) on the ‘emotion in music’ challenge using the DEAM dataset at the MediaEval Multimedia Evaluation Campaign. The performance of the best models in terms of RMSE is 0.08 (both for valence and arousal) [3]. The genre-aware linear model by Griffiths et al. [86] achieves an RMSE of 0.447 and 0.440 for valence and arousal, respectively. This confirms that the benchmark models proposed on our dataset can compete with state-of-the-art models.

### 6.2. Emotion Prediction Models That Use Audio as Well as Profile Info

In Table 6, the MSE and R-values for both types of models are shown for Valence and Arousal. Each row represents the results for the models trained with audio information and one additional profile feature as input. We should note that, as stated above, the datasets use averaged ratings per participant per profile feature, which results in differences in the dataset size depending on the number of feature bins. The last row of Table 6 corresponds to Table 5 and was included for convenience.

A graphical representation of Table 6 is depicted in Figure 7. The top two graphs show the MSE results, and we can observe that for valence, the fully connected model outperforms the LSTM model in all cases. For arousal, LSTM performed slightly better for models that included the age, genre, instrument, training, duration and master profile feature. Similarly, for the Pearson correlation (R) results, the fully connected model for valence outperforms the LSTM model in almost every case, except for the model that included the language profile feature. When predicting arousal, the LSTM model performs better for most models with profile features except those that included gender and residence.

We should note that it is not fair to directly compare the results of these different models with each other, as the test set is different for each of these models. For instance, in the case of the ‘instrument’ model, the model is not only trying to predict one valence and arousal value for each time step of the song, but one for participants who do not play an instrument, and one for those who play at least one instrument. This leads to less noisy labels overall, and a more directed prediction. Some profile features, like the duration of musical training, have bins with very few participants inside. In the case of musical training, there are three bins in total, with the >5 years bin containing only 0.9% of the data. Yet, due to the fact that the data is averaged per song per profile bin, this accounts for a third of the final model evaluation. So while some of the results with profile information may seem lower than the audio-only model, the predictions within certain bins will be stronger. We offer these results as a benchmark for future research, not so much to compare to the non-profile methods.

As seen in Section 4, there are significant differences in ratings between different groups. Based on the statistical analysis of the various profile groups in that section, the profile features related to musical background are related to VA ratings. We see that models that include those seem to have a higher predictive power, which is indicative of a better-separated feature space.

Table 6 offers benchmark prediction results for each of the models with different profile features as input. In future work, it would be interesting to explore the class-specific accuracy of each profile feature bin. Given the fact that we did always have a large set of ratings for each of the dedicated combinations of profile bins (e.g., Indian, with other language, more than 5 years of training, and other gender), we did not include a model that takes all profile features as input.

## 7. Conclusions

In this work, we present a new dataset of music with emotion ratings, called MERP, which includes continuous valence and arousal ratings for full-length audio, as well as profile information of the raters. On average, each song was rated by approximately 47 participants. We perform thorough data preprocessing on the dataset to clean it, including a novel approach for quality control based on benchmark ratings from the DEAM dataset. The resulting dataset is available online.

Through a detailed descriptive data analysis in Section 4, we uncovered which profile information has an influence on valence and arousal ratings. For instance, participants whose favourite genre is classical music, rate valence significantly different than participants with other favourite genres. For participants who reside in different countries (US versus India), we notice a significant difference in both arousal and valence ratings. This is in line with findings by Gómez Canón et al. [87], who state that profile information has the potential to improve group-based MER. We also found more differences between profile groups when taking into account culture-related profile types as well as music-related profile features.

We provide two baseline predictive emotion models for our new dataset based on a fully connected, as well as a long short-term memory neural network architecture. In an experiment, we examine the power of adding a profile feature as input to the model so as to get more customized ratings. Our proposed MERP dataset (https://www.kaggle.com/kohenyan/music-emotion-recognition-with-profile-information (26 December 2022)) as well as all of our baseline models (https://github.com/dorienh/MERP (26 December 2022)) are available online. We show that by providing not just denoised emotion ratings for full-length musical pieces, but also a set of profile features for each rater, we can use these data to train models that predict how specific groups of people perceive emotion. This will help address the noisiness that is inherent in the field of emotion ratings.

In future work, MERP can easily be expanded. The listening study can be opened again on MTurk, to the same regions for more data, or to more countries other than India and the US. More royalty-free music can be added as well. As long as the 4 DEAM songs are included in the study, they can continue to be used as a benchmark for identifying anomalies. We provide the code for the listening study (https://github.com/dorienh/MERP/blob/master/amazon_Merged.html) used on Amazon Turk as reference. Further, researchers have long investigated how particular music features convey emotional information. In a recent review, Panda et al. [12] survey emotion models using eight musical features—melody, harmony, rhythm, dynamics, tone colour, expressivity, texture and form—and discuss which aspects of these features may influence emotions. The dataset provided in this research provides an avenue for researchers to further examine musical features and their effect on the emotion states of listeners with different backgrounds. As the audio is royalty-free and available for download, future studies need not use the same openSMILE audio features used in this work, but can freely generate other features as well. The labels provided are dynamic and span full-length songs, and can be aggregated or split as desired. The labels are also 2-dimensional and can be mapped to categories according to hybrid emotional models such as [29].

In future research, it would be useful to further improve the baseline models with state-of-the-art machine learning techniques, and build a model that takes into account all features. It would also be useful to explore how emotion ratings typically evolve over the course of long music pieces. In order to account for subjectivity, not only for each song but also for each participant profile group, we prioritised collecting more labels for a small number of songs rather than fewer labels for many songs, while we have already collected numerous ratings for 50 full-length songs, in future research this could be further expanded. Having a larger variety of songs from more genres and styles would benefit the generalisation of predictive models.

## Figures and Tables

**Figure 1 sensors-23-00382-f001:**
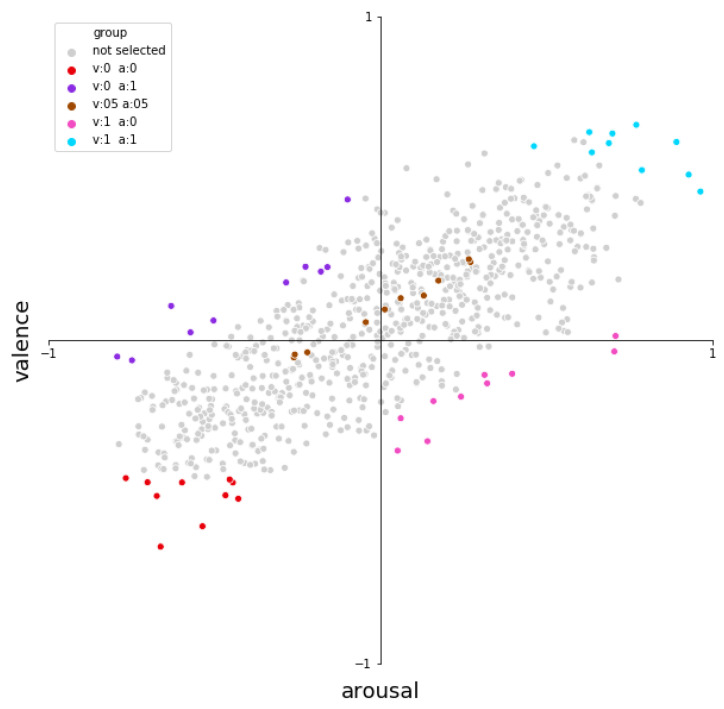
Representation of the predicted arousal and valence value of the top 1000 songs of the FMA (after filtering on length). The coloured dots represent the songs selected for each of the 5 categories for the final user study.

**Figure 2 sensors-23-00382-f002:**
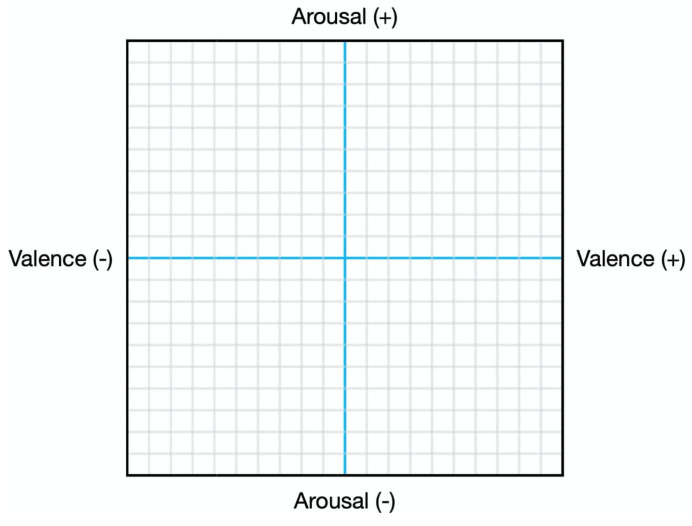
The interface through which valence and arousal values were captured via mouse tracking in the listening study.

**Figure 3 sensors-23-00382-f003:**
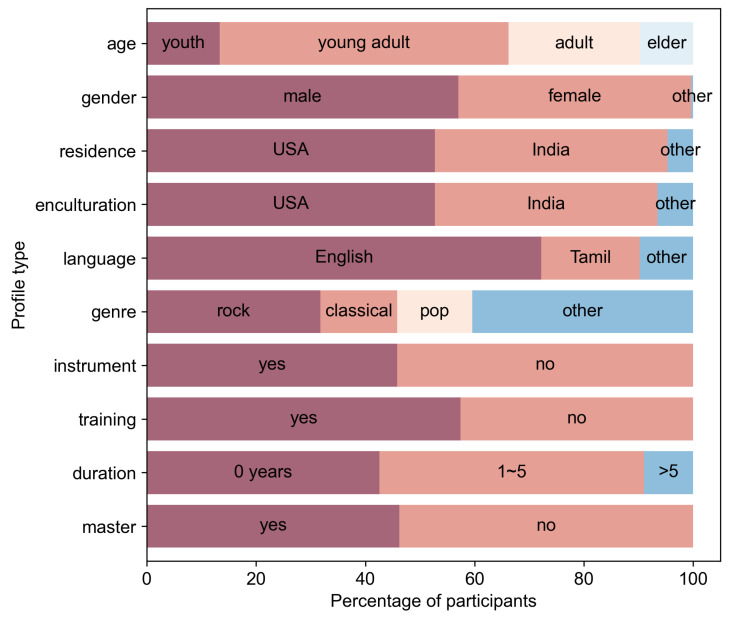
Proportion of participants for each of the profile features.

**Figure 4 sensors-23-00382-f004:**
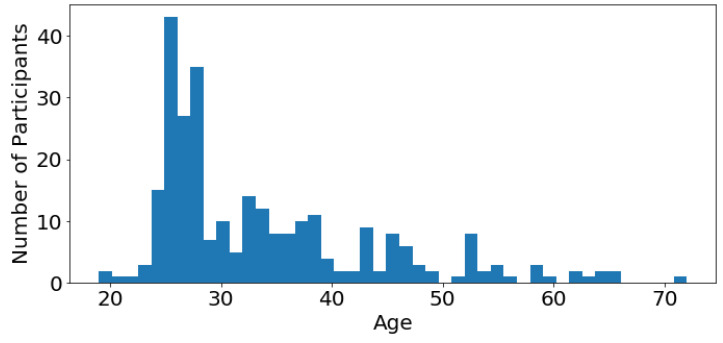
Age of participants.

**Figure 5 sensors-23-00382-f005:**
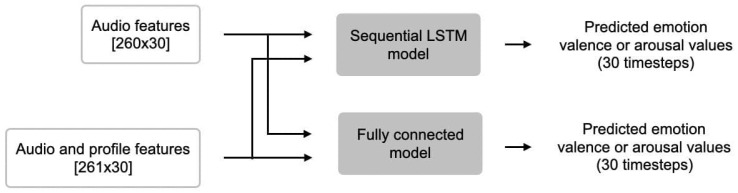
Overview of the two proposed models and their input/output.

**Figure 6 sensors-23-00382-f006:**
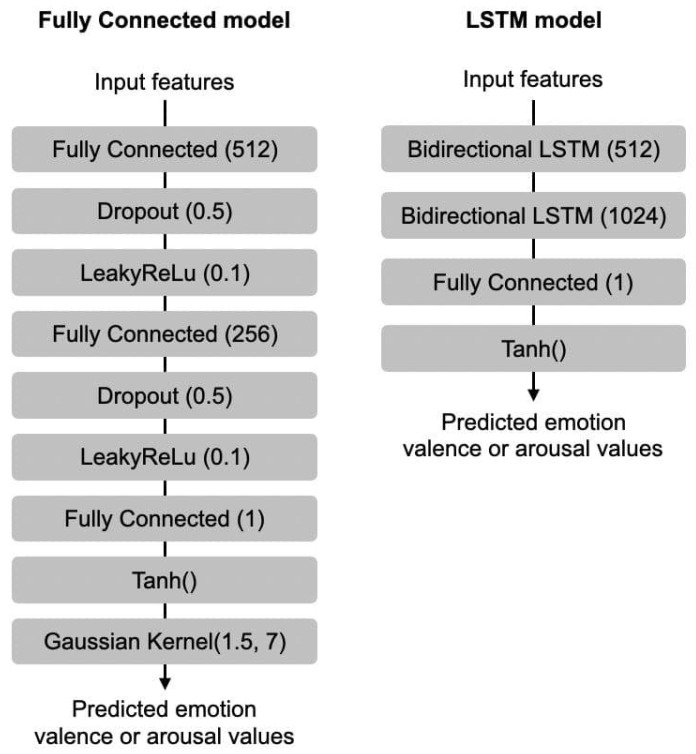
Baseline architectures. **Left**: Fully Connected architecture. **Right**: LSTM architecture.

**Figure 7 sensors-23-00382-f007:**
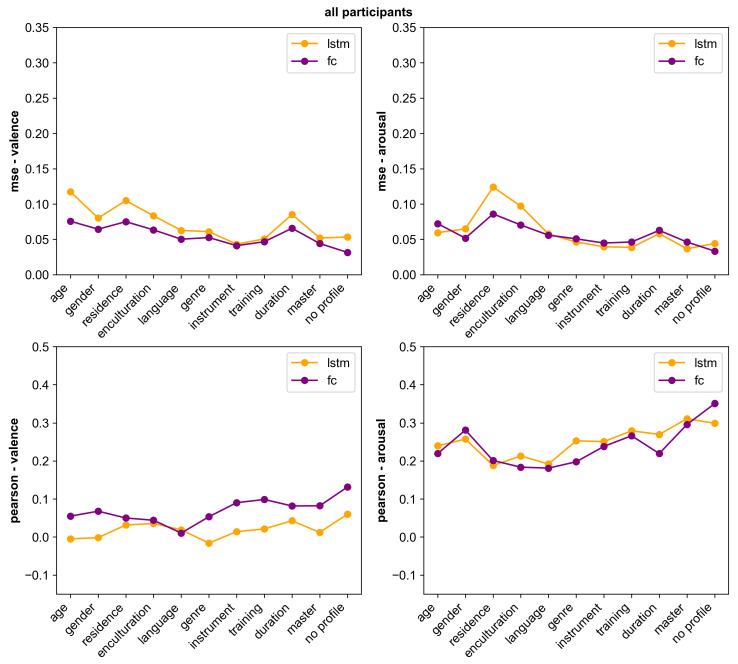
Comparison of evaluation metrics between the models trained using different profile features. Note that the range of the y-axes of the graphs has been adjusted for viewing convenience.

**Table 1 sensors-23-00382-t001:** Total duration of all of the stimuli for each Arousal-Valence category (mins).

Valence	Arousal	Minutes
low	low	34.1
low	high	23.0
mid	mid	31.5
high	low	16.0
high	high	38.7

**Table 2 sensors-23-00382-t002:** Overview of the participants and trials in dataset.

	Raw Data	
Participant Type	Number of Participants	Number of Trials
Master	139	1722
Non Master	219	2719
Total	358	4441
	**After Preprocessing**	
**Participant Type**	**Number of Participants**	**Number of Trials**
Master	128	1588
Non Master	149	1894
Total	277	3482

**Table 3 sensors-23-00382-t003:** Resulting *p*-values from the Kruskal Wallis tests run on each profile feature with valence and arousal ratings, respectively. Values in bold are statistically significant.

Profile Type	Valence *p*-Value	Arousal *p*-Value
age	**0.0191**	0.4907
gender	0.2166	0.1851
residence	**0.0156**	**0.0125**
enculturation	**0.0000**	**0.0198**
language	0.4050	0.1729
genre	0.0767	**0.0001**
instrument	**0.0002**	**0.0383**
training	**0.0009**	**0.0313**
duration	**0.0034**	**0.0022**
master	0.5507	**0.0015**

**Table 4 sensors-23-00382-t004:** List of 260 features extracted when using the configuration file IS13_ComParE_lld-func.conf from openSMILE. For a detailed description of these features, the reader is referred to [80].

Feature Name	Size
F0final	4
audSpec_Rfilt	104
audspecRasta_lengthL1norm	4
audspec_lengthL1norm	4
jitterDDP	4
jitterLocal	4
logHNR	4
pcm_RMSenergy	4
pcm_fftMag_fband1000-4000	4
pcm_fftMag_fband250-650	4
pcm_fftMag_mfcc	56
pcm_fftMag_psySharpness	4
pcm_fftMag_spectralCentroid	4
pcm_fftMag_spectralEntropy	4
pcm_fftMag_spectralFlux	4
pcm_fftMag_spectralHarmonicity	4
pcm_fftMag_spectralKurtosis	4
pcm_fftMag_spectralRollOff25.0	4
pcm_fftMag_spectralRollOff50.0	4
pcm_fftMag_spectralRollOff75.0	4
pcm_fftMag_spectralRollOff90.0	4
pcm_fftMag_spectralSkewness	4
pcm_fftMag_spectralSlope	4
pcm_fftMag_spectralVariance	4
pcm_zcr	4
shimmerLocal	4

**Table 5 sensors-23-00382-t005:** Model performance when using only audio features as input. Best values in bold.

	Valence		Arousal	
	MSE	R	MSE	R
Fully connected	**0.0315**	**0.1314**	**0.0333**	**0.3507**
LSTM	0.0532	0.0599	0.0441	0.2992

**Table 6 sensors-23-00382-t006:** Model performance when using both audio features as well as one profile feature as input. The best performing model is indicated in bold.

	Fully Connected				LSTM			
Profile Feature	Valence		Arousal		Valence		Arousal	
	MSE	R	MSE	R	MSE	R	MSE	R
age	0.0756	0.0549	0.0722	0.2194	0.1174	−0.0048	0.0595	0.2397
gender	0.0644	0.0679	0.0517	0.2813	0.0802	−0.0017	0.0649	0.2576
residence	0.0750	0.0500	0.0860	0.2011	0.1048	0.0316	0.1240	0.1881
enculturation	0.0634	0.0441	0.0703	0.1837	0.0834	0.0355	0.0970	0.2133
language	0.0502	0.0104	0.0559	0.1812	0.0626	0.0189	0.0576	0.1919
genre	0.0525	0.0534	0.0507	0.1978	0.0609	−0.0160	0.0462	0.2530
instrument	**0.0411**	0.0903	**0.0448**	0.2383	**0.0432**	0.0142	0.0394	0.2509
training	0.0466	**0.0987**	0.0462	0.2663	0.0506	0.0214	0.0386	0.2793
duration	0.0657	0.0817	0.0627	0.2195	0.0853	**0.0427**	0.0580	0.2698
master	0.0441	0.0821	0.0462	**0.296**	0.0519	0.0122	**0.0366**	**0.3110**
audio only	**0.0315**	**0.1314**	**0.0333**	**0.3507**	0.0532	**0.0599**	0.0441	0.2992

## Data Availability

The dataset as well as the model code are available online at http://www.kaggle.com/kohenyan/music-emotion-recognition-with-profile-information and https://github.com/dorienh/MERP, respectively. (accessed on 26 December 2022).

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
