# Peer review of "MERP: A Music Dataset with Emotion Ratings and Raters’ Profile Information"

_sensors, 2022, doi:10.3390/s23010382_

Round 1
Reviewer 1 Report
The paper presents a new dataset of music with emotion ratings, called MERP, which includes continuous valence and arousal rating for full-length audio, as well as profile information of the raters.
Russell's emotional valence-arousal model should be included in the paper.
The models presented in sections 4.2.1 and 4.2.2 should be better explained, indicating the reason for the parameters.
The paper would be improved by including a comparison with other current music databases where emotions are also detected.
In the bibliographic reference
Wang, X.; Wei, Y.; Heng, L.; McAdams, S. A Cross-cultural Analysis of the Influence of Timbre on Affect Perception in Western Classical Music and Chinese Music Traditions. Frontiers in Psychology, p. 4272.
the year is missing.
Reviewer 2 Report
The paper is well written and clear. Authors provides a useful dataset for the research community. However, the benchmark classifiers could be improved. The benchmark classifiers seems to not be benefited by the profile information.
Reviewer 3 Report
Title -----------------------------------
I recommend to change the paper's name. It seems a bit confuse.
L09: Correct the text to '..DEAM dataset..'
Introduction -----------------------------------
The introduction presents so many historic context of the song and emotion. maybe it can be better resumed and more direct related to the present work.
Data gathering procedure -----------------------------------
L181: Please, REMOVE all personal pronouns/possessives (we, you, our, my, I, they, their, etc).
L182: Please write 'A total of 182 participants were considered' instead of 'There are two types 182 of participants'
L183: The use of apostrophe is incorrect along the text. Use ` and ´, not ´ and ´.
L103: Rewrite. '..DEAM, which IT contains...' . I saw other parts having the same mistake. Please. Review the text on that.
L211: Remove 'we' and all personal pronouns along the text.
Figure 1 must to be larger and and the scatter plot needs to present better the dots. The dots are really tiny.
Dataset analysis and visualisation -----------------------------------
L280: Remove 'we' and all personal pronouns along the text.
L280: How many participants the work really used? The abstract says 277. Alongn the text we can see other number e.g., 452, 447 or 277participants. Mybe it must to be clear in the abstract too.
L352: The use of apostrophe is incorrect along the text. Use ` and ´, not ´ and ´. Check it along all text.
Emotion prediction model -----------------------------------
Opensmile is a huge tool to do analysis of speech, audio and other stuff on frequency/spectral analysis and features straction, but the paper needs to detailed the features better. To present a Table detailing these features, dividing them by type, e.g., for time, frequency, banks, and so on.
L491: Please, put a Table, describing all the features that were used in the model. MFCC was considered? What are the other spectral features? Statistical features were also considered? FFT?SFFT? Wavelets? The reduction of dimensionality of the Features was used e.g., PLA, SVD? What were used. The authors need to present it in details to really improve the science and contribute to the science. I mean, what kind of features?
L549-550: 'LSTMs are a type of Recurrent Neural Network (RNN), which IT are designed for sequential data'. Actually, this isn't a so correct definition of the powerful LSTM. It is a architecture of some RNN, having feedbacks and it is useful to random data too not only for sequential data; it is another option unlike standard feedforward neural networks. Also, 'sequential data' is really technical word on the context of sound,video, speech that, I thing, the authors thought to refer to it. So, I recomend the authors to use 'sequential data', but putting between brackets 'speech or video....', i.e., sequential data (video, speech..........).
Please, provide a better definition of the LSTM, and consider wat I recommend. It is really important to be present.
Results -----------------------------------
Table 4, needs to be better explained its results. In two situations, the MSE was bigger using LSTM than Fully conn. What is the true reason for that? To me, according to what was presented on Fully conn, it makes no sense, since I know the reduction of the hidden layers. Another question: the authors used sigmoid function to compare or only tanh( )? On prediction context and in several papers, sigmoid/logistic presents really good results too as activation functions.
References -----------------------------------
The close to 87,2% of allreferences, are from 4 ou 5 years ago. It makes no sense, since today we have several actual researches that find different solutions in similar scope of the present work. Please, revise it!
FINAL REMARKS -----------------------------------
In general, the presented journal presents good explanation of the problem, having a good number of volunteers in experiment. Unhappiness, since we use data analysis I considers mandatody to present more details regarding to activation function comparison and the features used and why used them; good plots and scatterplots is important to have more some; it is fundamental to analyze cross correlation of the data in scene.
To conclude. The paper is good, but I will considers it only after attend all points I suggest, because there are things to be solved in the papers. For now, I will put as Reconsider after major revision. Once all points be done, I accepted!
Best regards!
